

# Filtered mud improves sugarcane growth and modifies the functional abundance and structure of soil microbial populations

Ahmad Yusuf Abubakar[1,2,3], Muhammed Mustapha Ibrahim[4,5], Caifang Zhang[1,2], Muhammad Tayyab[1,2], Nyumah Fallah[1,2], Ziqi Yang[1,2], Ziqin Pang[1,2] and Hua Zhang[1,2]

[1] College of Agriculture, Fujian Agriculture and Forestry University, Fuzhou, Fujian, China
[2] Key Laboratory of Sugarcane Biology and Genetic Breeding, Ministry of Agriculture, Fujian Agriculture and Forestry University, Fuzhou, Fujian, China
[3] Bioresources Development Centre, National Biotechnology Development Agency, Kano, Nigeria
[4] College of Resources and Environment, Fujian Agriculture and Forestry University, Fuzhou, Fujian, China
[5] Department of Soil Science, University of Agriculture Makurdi, Benue, Nigeria

Corresponding author
Hua Zhang,
zhanghua4553@sina.com

## ABSTRACT

**Background:** Exploring high-quality organic amendments has been a focus of sustainable agriculture. Filtered mud (FM), a sugar factory waste derived from sugarcane stems, could be an alternative organic amendment for sugarcane production. However, the effects of its application proportions on soil fertility, nutrient cycling, structure of soil bacterial and fungal communities, and the growth of sugarcane in clay-loam soils remain unexplored.

**Methods:** Three application proportions of FM: (FM1-(FM: Soil at 1:4), FM2-(FM: Soil at 2:3), and FM3-(FM: Soil at 3:2)) were evaluated on sugarcane growth and soil nutrient cycling. High throughput sequencing was also employed to explore soil microbial dynamics.

**Results:** We observed that FM generally increased the soil's nutritional properties while improving $NO_3^-$ retention compared to the control, resulting in increased growth parameters of sugarcane. Specifically, FM1 increased the concentration of $NH_4^+-N$, the N fraction preferably taken up by sugarcane, which was associated with an increase in the plant height, and more improved growth properties, among other treatments. An increase in the proportion of FM also increased the activity of soil nutrient cycling enzymes; urease, phosphatase, and β-glucosidase. High throughput sequencing revealed that FM reduced the diversity of soil bacteria while having insignificant effects on fungal diversity. Although increasing FM rates reduced the relative abundance of the phyla *Proteobacteria*, its class members, the *Gammaproteobacteria* and *Betaproteobacteria* containing some N-cycling related genera, were stimulated. Also, FM stimulated the abundance of beneficial and lignocellulose degrading organisms. These included the bacterial phyla *Actinobacteria*, *Bacteroidetes*, *Acidobacteria*, *Chloroflexi*, and the fungal phylum *Ascomycota*. The distribution of the soil microbial community under FM rates was regulated by the changes in soil pH and the availability of soil nutrients. Since FM1 showed more promise in improving the growth properties of sugarcane, it could be more economical and sustainable for sugarcane production in clay-loam soils.

## INTRODUCTION

Sugarcane (*Saccharum officinarum* L.) is a globally important crop that contributes significantly to the raw materials needed in sugar and biofuel-producing industries (*Khalil et al., 2018*). It is cultivated in the tropical and subtropical regions of the world and has an annual output of approximately 16 million tons (*Fallah et al., 2021*). Over the years, there has been extensive use of inorganic fertilizers, especially nitrogen (N), in sugarcane production to meet its increasing industrial demands (*Azeem et al., 2014*; *Thorburn et al., 2017*). However, the excessive application of these chemical fertilizers does not always show a continuous positive effect on crop production (*Vitousek et al., 2009*; *Bei et al., 2018*). Instead, it can result in low nutrient use efficiency and soil acidification (*Bei et al., 2018*), eutrophication and leaching (*Ibrahim et al., 2020a*), N deposition, as well as greenhouse gas emissions (*Liu et al., 2013*). Besides, the prolonged application of inorganic fertilizers could result in changes in the physical, chemical, and biological properties of the Soil (*Rivera-Becerril et al., 2017*). Alternatively, the use of organic fertilization to mitigate soil acidification and improve soil nutrient status, thus ensuring sugarcane productivity, is a promising approach. Organic amendments such as farmyard manure and crop residues have been shown to improve soil physicochemical properties, productivity, microbial community diversity, and composition compared to chemical fertilizers (*Cesarano et al., 2017*; *Das et al., 2017*). Therefore, the continuous exploration of several high-quality organic materials would also be beneficial in improving the soil's physical, chemical, and biological properties for crop production.

Filtered mud (FM) is a solid waste by-product produced from sugar mills during sugar production. It comprises about 3% of the crushed sugarcane after sulphonation (*Kumar et al., 2011*) and 7% after carbonation (*Saleh-e-In et al., 2012*). This results in a dark brown amorphous and soft solid comprising sugar, fiber, coagulated colloids, soil particles, inorganic salts, and several mineral elements in variable proportions (*Joshi, Sharma & Kangri, 2010*). Studies have shown that FM could effectively increase soil pH, prevent soil erosion, crusting, cracking, improving drainage, and stimulate soil microbial abundance (*Tandon, 1995*). Its sole or combined application with chemical fertilizer has been shown to increase the organic carbon (C), total nitrogen (N), phosphorus (P), and potassium (K) status of the Soil (*Kaur, Kapoor & Gupta, 2005*). However, management systems utilizing organic amendments could have varying effects in stimulating the activity and diversity of soil microbial community and the nutrient content of the soil depending on their source and composition (*Zhang et al., 2015*). Unfortunately, there is limited information on the effects of FM on the changes in soil nutritional composition through its alteration of the diversity and composition of soil bacterial and fungal communities. Therefore, understanding the impact of different organic materials used in soil management on the shifts in soil microbial diversity and community abundance could help us understand soil nutrient cycling processes (*Dai et al., 2017*).

While several organic amendments have been evaluated for their potential in restoring soil fertility and productivity, the evaluation of FM in clay-loam soils under sugarcane cultivation has not been documented. Adequate information on the effect of FM on soil quality, plant growth, and soil bacterial and fungal diversity remains relatively scarce in soils having high water retention capacity. Moreover, how different FM amendment proportions could alter the diversity and structure of soil microbial populations in the rhizosphere of sugarcane crops in such soils has not been previously reported. Also, the shift in fungal community composition may be influenced by the use of FM, as there has been evidence of fungal composition in FM (*Tayyab et al., 2019*). This could alter the general soil microbial diversity and balance when incorporated into the soil. Similarly, the relationship between soil bacterial and fungal diversity and structure with soil properties due to different FM rates requires adequate understanding. These, therefore, indicate significant knowledge gaps in understanding the effects of FM and its application rates on soil microbial diversity and functional abundance. Therefore, evaluating soil microbial dynamics using high-throughput sequencing will be helpful to reveal the complexity and diversity of microbial communities (*Shendure & Ji, 2008*; *Dai et al., 2017*).

We hypothesize that different FM:soil ratios would have varying effects on the availability of soil nutrients, sugarcane growth, nutrient cycling enzyme activity, microbial diversity, and community structure in clay-loam soil. Therefore, the objectives of this study were; (a) to evaluate the effect of FM rates on soil nutrient supply potential and agronomic characteristics of sugarcane plants in clay-loam soil, (b) to investigate the diversity and relative abundance of soil bacterial and fungal communities after FM amendment, (c) to investigate the roles of soil properties in the dynamics of the bacterial and fungal community composition of the rhizosphere of sugarcane in a clay-loam soil amended with different FM proportions.

## MATERIALS AND METHODS

### Soil substrate preparation and experimental design

The soil used for the experiment was collected from the Fujian Agriculture and Forestry University sugarcane cultivation field (latitude: 26°05 9.60″N; longitude: 119°14 3.60″E). The physicochemical properties of soil were; total carbon (TC): 0.97%, total phosphorus (TP): 0.7 g/kg, and total nitrogen (TN): 0.10%. The soil particle size distribution was clay: 20.3% silt: 43.1% sand: 36.6%, which was classified as clay-loam. The sugar mill filtered mud (FM) used in this experiment was obtained from Nanjing Qinfeng Crop Straw Technology Company, China. Before mixing with the soil, the FM was oven-dried at 50 °C until a constant weight was achieved. The chemical properties of the FM are as follows; pH: 7.28, EC: 3.45 dS/m, O.C: 23.31%, O.M: 40.1%, N: 1.64%, *P*: 14.4 g/kg, K: 14.1 g/kg, Ca: 1.33 cmol/kg, Mg: 1.37 cmol/kg, S: 1.52%, and Na: 1.44 cmol/kg.

The experiment was carried out in a greenhouse at the Fujian Agriculture and Forestry University Fuzhou, Fujian Province, P.R China, from March to December 2019 using red PVC pots. Each pot had a height of 180 mm and a diameter of 120 mm. The experimental soil was properly air-dried and sieved into a 2 mm size fraction and mixed according to the respective ratios with FM to give a total weight of 10 kg per pot.

The treatments evaluated comprised of a control (CK) without any amendment and three different FM:soil ratios. These FM ratios were hypothetically selected to accommodate its possible proportions that can be used under various conditions, from fields requiring more soil to greenhouse cultivation requiring less soil. These treatments were: CK, FM1 (FM:soil at 1:4), FM2 (FM:soil at 2:3), and FM3 (FM:soil at 3:2). The treatments were replicated three times and arranged in a randomized complete block design.

After mixing the soil and FM according to the respective proportions, the pots were watered to field capacity and left for 24 h to equilibrate. Sugarcane stems were planted vertically in each pot and periodically irrigated to field capacity using tap water. Each pot was also kept weed-free by hand weeding throughout the experimental period. During the experiment, the air temperature recorded was 25–27 °C, and relative humidity of 75–80% was maintained throughout the experiment. The agronomic characteristics (as listed in the plant data collection section) of the sugarcane plants were recorded at the tillering stage, which signified the period of rapid nutrient uptake for reproductive growth.

At the plant's tillering stage (90 days after planting), about 40 g of soil sample was collected from each pot's root zone using a portable soil auger of 1 cm diameter. Six subsamples were taken around and within the root zone of the crop per pot up to a depth of 18 cm to represent the rhizosphere soil and bulked to form a composite. A subsample of fresh soil was taken and put in well-labeled bags and stored in a refrigerator at 4 °C for the evaluation of the soil enzyme activity. Part of the subsampled soil was air-dried, ground, and sieved using a 2-mm sieve to analyze the soil's physicochemical properties. About 10 g of the fresh soil subsamples were collected and stored at −20 °C for DNA extraction.

## Chemical properties of the soil

The pH of the soil samples was measured with a glass electrode pH meter using 1:2.5 (weight/volume) soil:water ratio (*Ibrahim et al., 2020b*). Electrical conductivity (EC) was measured with a conductivity meter, while the soil total C and N were measured using the Flash Smart elemental analyzer (Thermo Scientific™, Waltham, MA, USA). The soil available K was determined by extraction using the ammonium acetate method and measured using flame photometry (*Pansu & Gautheyrou, 2007*). The molybdenum blue method was used to measure the available phosphorus (AP). Soil ammonium ($NH_4^+$) and nitrate ($NO_3^-$) were extracted using 2 M KCl, and measured by a Bran+Luebbe GmbH-AutoAnalyzer 3 (Bran+Luebbe, Norderstedt, Germany). The activities of the soil urease, phosphatase, β-Glucosidase, and cellulase were measured as described by *Sun et al. (2014)*. Soil urease activity assay was based on the $NH_4^+$−N released when the samples were incubated with 10 mL of 10% urea solution and 20 mL of citric acid buffer (pH 6.7) at 37 °C for 24 h. Cellulase activity assay involved the determination of reducing sugars produced when the soil sample was incubated with acetate buffer (50 mM, pH 5.5), carboxymethyl cellulose, and toluene at 37 °C for 24 h (*Deng & Tabatabai, 1994*). Soil phosphatase activity was determined after incubating the samples with 0.25 mL of toluene and 1 mL of disodium p-nitrophenyl phosphate tetrahydrate and placed in a water bath for 1 h at 37 °C (*Tabatabai & Bremner, 1969*). The assay for the determination of the

β-glucosidase enzyme activity was carried out after incubating the samples with 50 mM cellobiose substrate solution in citrate–phosphate buffer (pH 6.30) in a shaker at 37 °C for 1 h (*Stege et al., 2010*).

## Plant data collection

The plants' height (PLH) and diameter (PLD) were measured with the aid of a meter rule and Vernier caliper, respectively. The PLH in each pot was measured from the base of the plant to the top visible dewlap (TVD) leaf. The Vanier caliper was used to measure the stalk diameter around the middle of the stalk. The number of leaves (NL) and the number of tillers (NT) on each plant were counted and averaged at the tillering stage. In addition, the leaves' chlorophyll content (Chol) was measured using a chlorophyll meter (SPAD-502; Minolta, Osaka, Japan).

## Soil genomic DNA extraction and amplicon sequencing

The total genomic DNA of the soil was extracted from 0.5 g of each soil sample using the Fast DNA$^{TM}$ Spin kit for soil following the manufacturer's instructions (MP Biomedical, Solon, OH, USA). The extracted soil DNA's quality was visualized on 1% gel electrophoresis and assessed using a NanoDrop ND-2000 spectrophotometer (NanoDrop Technologies, Thermo Scientific, Wilmington, DE, USA). The DNA concentration was thereafter quantified using the Qubit assay (Invitrogen, CA, USA).

The amplification of the v3–v4 region of the 16S rRNA genes in the extracted DNA as well as the v4 region of 18S rRNA were carried out using the polymerase chain reaction (PCR). The 341F/797R forward and reverse primers with barcodes were used for the PCR amplification of the 16S rRNA genes (*Caporaso et al., 2010*), while the 1176F/1536R were used for that of the 18S rRNA genes (*Smit et al., 1999*). The PCR amplification was carried out on each samples using three replicates, with a mixture containing 4 μL of 5 × Reaction Buffer, 2 μL of dNTPs (2.5 mM), 0.25 μL of TransStart Fastpfu DNA Polymerase, 1 μL of each primer (5 μM) (reverse and forward), and ~10 ng template DNA (*Michelsen et al., 2014*). The PCR conditions involved an initial denaturation of 98 °C for 1 min, followed by 30 cycles of denaturation at 98 °C for 10 s, annealing at 50 °C for 30 s, extension at 72 °C for 60 s, and a final extension at 72 °C for 5 min. A 2% agarose gel electrophoresis was carried out to visualize the amplified DNA. The amplified PCR products were thereafter recovered by cutting the gel using the AxyPrep DNA Gel Recovery Kit (Axygen Biosciences, CA, USA). Samples that are characterized by one bright main strip between 400 bp and 450 bp were selected for further sequencing. The purification of the obtained PCR product was subsequently done using a Gel Extraction kit (TIANGEN, Beijing, China). An equal concentration of the purified amplicons was mixed into a single tube for sequencing. The initiation of the sequencing libraries was carried out in Illumina with the aid of a specific NEB Next® Ultra$^{TM}$ DNA Library Prep Kit. The library sequencing quality having an average insert size of 400 bp was used for paired-end sequencing on an Illumina MiSeq platform at the New England Biolabs Ltd. Beijing, China. The raw reads generated from amplicon sequencing were submitted to the Sequence Read Archive of the NCBI (https://trace.ncbi.nlm.nih.gov/Traces/sra/) under

the bioProject PRJNA755433, and biosample accessions from SAMN20822645 to SAMN20822648 for soil bacteria, and SAMN20822649 to SAMN20822652 for the fungi.

## Statistical analysis and bioinformatics

Data obtained from plant characteristics and soil physicochemical properties were analyzed using the analysis of variance (ANOVA), and the means obtained were separated using Tukey's test at $p < 0.05$.

Quantitative Insights into Microbial Ecology (QIIME2; https://qiime2.org) pipeline (*Bolyen et al., 2019*) was used to classify raw sequences according to the specific barcode assigned to each sample. The original DNA segments were combined with the Paired-end reads (PERs), using FLASH (Baltimore, MD, USA) (*Magoč & Salzberg, 2011*). PERs were designed for each sample according to the particular barcodes connected with DNA fragments. UPARSE-OTU reference algorithm was used to analyze the sequences with the UPARSE software package (CA, USA). Truncation and removal of chimeric sequences were done using the UCHIME algorithm. It was ensured that the average quality score remained above 33 with regards to the 50 bp sliding window when calculating the truncation thresholds. The clean tags obtained were clustered to generate operational taxonomic units (OTUs) at a 97% similarity. The taxonomic identities of the soil microbial communities were checked using the Silva database (Release 138, http://www.arb-silva.de). The diversity and composition of the bacterial and fungal communities under different treatments were determined based on the instructions *Caporaso et al. (2010)*. The Chao1 and ACE (species abundance), as well as the Shannon and Simpson indices (community diversity), were used to estimate the fungal and bacterial alpha diversity (within samples). Krona charts were used to show the abundances of each bacterial and fungal taxa graphically. For Beta diversity, R (Version 3.3.1) was used to visualize the similarity distance using Principal Coordinate Analysis (PCoA). Selected soil physicochemical properties were used to establish the identity and the association of the most abundant bacterial and fungal phylum using partial-RDA.

## RESULTS

### Soil physicochemical characteristics

All the treatments containing FM, irrespective of its proportion, had significantly higher values ($p < 0.05$) of the measured soil chemical properties. There was no significant difference in $NH_4^+$ concentration among the FM2 and FM3. However, FM1 had a significantly higher $NH_4^+$ concentration among all the treatments evaluated. $NO_3^-$ had lower values in the FM treatments when compared to the control (CK) (Table 1). While FM increased the soils' pH relative to CK, there was no significant change in the soil pH among the FM treatments. Also, the highest AK, EC, TN, TON, and TOC values were obtained in FM3. Also, the AK, TOC, and TON values were significantly highest and statistically similar among FM2 and FM3 treatments. Additionally, the available P content was statistically higher in the FM treatments compared to the CK, although it was highest in value in the FM2.

**Table 1 Soil physicochemical properties at sampling.**

| Treatment | pH | EC (dSm-1) | AK (mg/kg) | AP (mg/kg) | TC (g/kg) | TN (g/kg) | TOC (mg/kg) | TON (mg/kg) | C/N | $NH_4^+$ (mg/kg) | $NO_3^-$ (mg/kg) |
|---|---|---|---|---|---|---|---|---|---|---|---|
| CK | 6.1± 0.13b | 25.28± 1.17d | 74.3± 1.5c | 7.4± 0.1d | 10.3± 0.4c | 1.5± 0.05d | 134.7± 5.9c | 171.1± 13.2c | 7.1± 0.06a | 10.2± 6.8b | 16.8± 1.1a |
| FM1 | 7.2± 0.01a | 209.9± 2.58c | 120.2± 1.6b | 57.0± 1.3b | 25.9± 1.2c | 6.3± 0.1c | 170.8± 7.1b | 201.7± 23.9b | 4.1± 0.2b | 16.6± 1.1a | 11.1± 0.5c |
| FM2 | 7.2± 0.08a | 290.7± 5.7b | 255.7± 1.5a | 65.7± 2.2a | 34.4± 1.7a | 8.2± 0.7b | 288.0± 14.7a | 581.1± 31.6a | 4.2± 0.5b | 10.7± 0.4b | 11.6± 0.4c |
| FM3 | 7.2± 0.06a | 376.7± 5.6a | 252.3± 3.1a | 47.9± 1.6c | 35.7± 0.5a | 9.6± 0.6a | 286.9± 13.3a | 574.0± 29.0a | 3.7± 0.3bc | 11.3± 0.3b | 14.3± 0.4b |

Notes:
CK (control), FM: Filtered mud, FM1 (FM:soil at 1:4), FM2 (FM:soil at 2:3), FM3 (FM:soil at 3:2);.
Means are followed by ± standard deviation (SD) ($n$ = 3).
Letters show significant differences between means ($p > 0.05$).
EC, Electrical conductivity; AK, Available potassium; AP, Available phosphorous; TC, Total carbon; TN, Total nitrogen; TOC, Total organic carbon; TON, Total organic nitrogen.

## Soil enzyme activity

The influence of FM proportion on the soil enzyme activity: cellulase, phosphatase, urease, and β-Glucosidase is presented in Figs. 1A–1D. Our observations showed that there was no significant ($p < 0.05$) effect of FM rates on cellulase activity (Fig. 1A). However, the control treatment had a significantly higher cellulase activity than the FM1 and FM2 while being statistically similar to FM3. On the other hand, phosphatase, urease, and β-Glucosidase enzymes were stimulated by FM relative to the control. Among the FM proportions, FM3 gave the highest ($p < 0.05$) activity of phosphatase and urease (Figs. 1B, 1C). While there was no significant difference in the activity of β-Glucosidase among the FM proportions evaluated, its activity was higher in the FM amendments than the control (Fig. 1D).

## Sugarcane agronomic characteristics

The application of FM, irrespective of its proportion, significantly increased the plant height, number of tillers, and plant stem diameter compared to the control treatment (Figs. 2A, 2D and 2E). The use of FM1 gave the significantly highest ($p < 0.05$) plant height, among other FM treatments. Similarly, FM1, while statistically similar ($p < 0.05$) to other FM treatments, gave higher values for the number of tillers (Fig. 2D) and plant diameter (Fig. 2E). However, FM treatments had no significant difference ($p < 0.05$) when compared to the control with respect to the leaves' chlorophyll content (Fig. 2B) and the number of leaves (Fig. 2C). There was no statistical difference in the leaves' chlorophyll content and the number of leaves among the FM treatments, despite the observed reduction in FM2 and FM3.

## Relative abundance of dominant microbial phyla in filtered mud amended soil

The relative abundances of the top 10 soil bacterial and fungal phyla as influenced by the different FM:soil ratios are presented in Figs. 3A, 3B. The *Proteobacteria* was dominant in

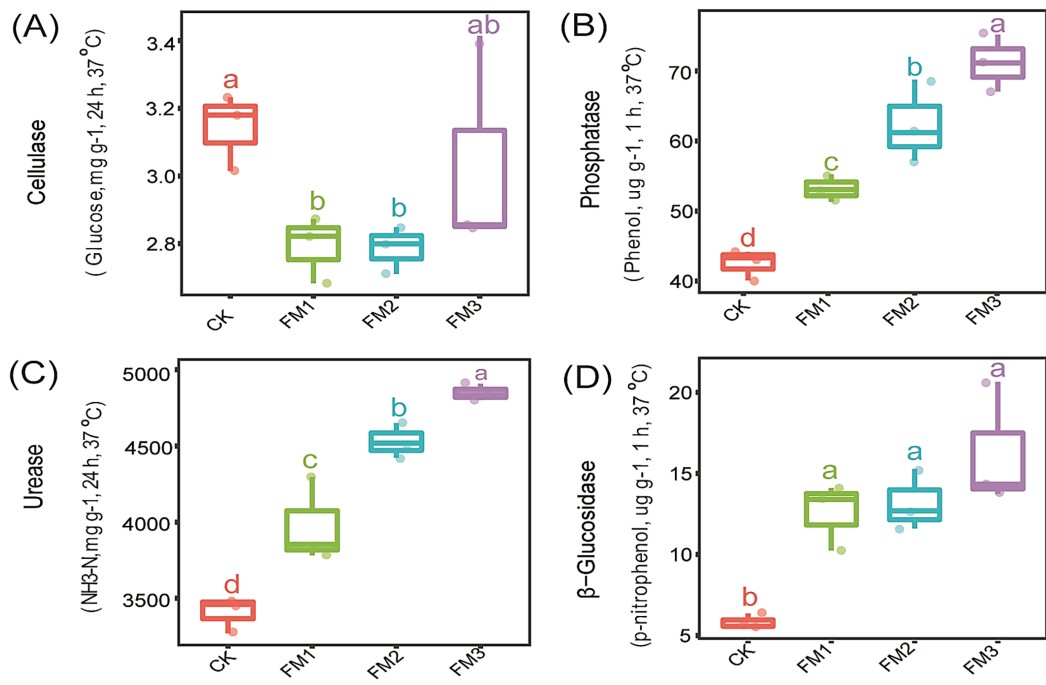

**Figure 1 Effects of filtered mud on soil enzyme activities (A) Cellulase (B) Phosphatase (C) Urease (D) β-glucosidase. CK (control), FM: Filtered mud; FM1 (FM:soil at 1:4), FM2 (FM:soil at 2:3), FM3 (FM:soil at 3:2).** The lowercase letters on the boxplots show significant differences between treatment means (Tukey's test, $p < 0.05$).

the soil, and their relative abundance ranged from 42.54–50.76% and decreased with an increasing rate of FM. The *Actinobacteria* (14.74–25.59%), which were the second most abundant bacterial phyla, increased in the FM treated soil relative to the control. Its highest proportion was obtained in FM3. These were followed by the *Acidobacteria* (4.51–11.48%), which decreased in the FM treated soils. Among the phyla observed, *Bacteroidetes* (3.65–6.03%), *Chloroflexi* (2.7–4.71%), *Planctomycetes* (1.16–3.49%), and *Candidatus Saccharibacteria* (0.43–2.83%) increased in the FM treated soils compared to the control. However, the relative abundances of the *Gemmatimonadetes* (3.44–5.9%) and *Firmicutes* (1.52–2.27%) also reduced in the FM treated soils compared to the control (Fig. 3A). Similarly, there was an alteration in the relative abundances of fungal phyla after the FM application (Fig. 3B). The most dominant fungal phyla observed were the *Ascomycota* (29.73–59.13%), whose relative abundance was reduced in the FM treated soils compared to the CK treatment. Its lowest abundance was observed in FM2, among the FM-treated soils.

The *Alphaproteobacteria* class dominated the *Proteobacteria* class members and was reduced in FM treatments relative to the control. Its lowest proportion was observed in FM3 (Fig. 4A). However, there was a notable increase in the relative abundance of the *Betaproteobacteria* and *Gammaproteobacteria*, which comprise members that are ammonium oxidizing due to FM. This was confirmed in the increase in the ammonium oxidizing genera, *Devoisa, Luteimonas*, and *Povalibacter* in the FM treatments (Fig. S1). Similarly, bacterial class distribution revealed the abundance of *Actinobacteria*, which contains N-cycling genera, such as the *Nocardioides* in FM treatments (Fig. S1), with FM 3

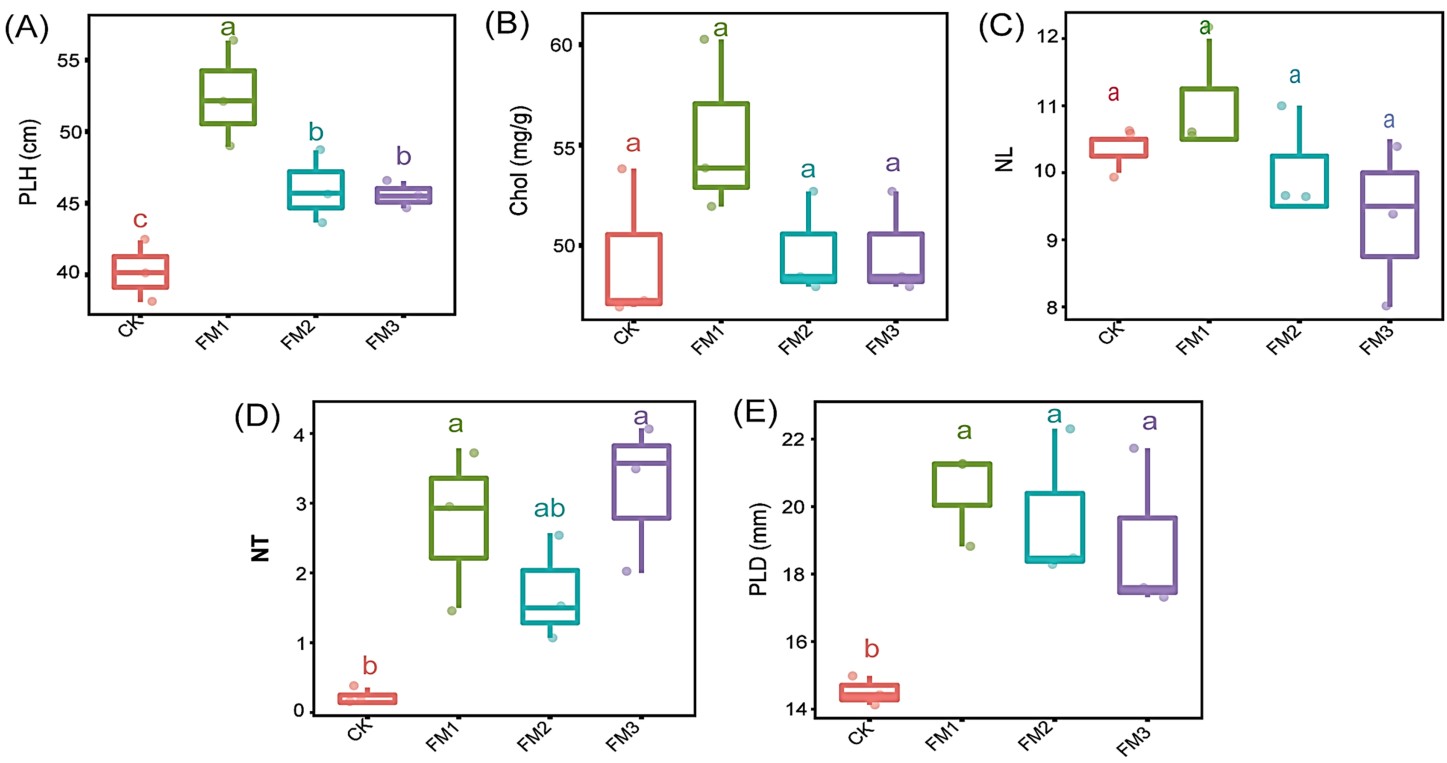

**Figure 2 Agronomic characteristics of sugarcane. (A) PLH, plant height; (B) Chol, Leaves chlorophyll content; (C) NL, Number of leaves; (D) NT, Number of tillers; (E) PLD, Plant diameter.** The lowercase letter on the boxplots show significant differences between treatment means (Tukey's test, $p < 0.05$). CK (control), FM: Filtered mud; FM1 (FM:soil at 1:4), FM2 (FM:soil at 2:3), FM3 (FM:soil at 3:2).

having the highest relative abundance (Fig. 4A). *Gemmatimonadetes* and *Sphingobacteriia* reduced in abundance in FM treatments relative to the control. The FM amendment suppressed the abundance of the fungal class *Sordariomycetes* and *Eurotiomycetes* (Fig. 4B). While the *Sordariomycetes* had their highest proportion in the FM2 treatment, there was no significant difference in the relative abundance of *Eurotiomycetes* among the FM treatments. The fungal class *Dothideomycetes* was, however, stimulated in FM, especially in FM1 and FM2. The *Saccharomycetes* were increased under FM, with their highest proportion in FM2.

## Alpha and beta diversity of soil bacterial and fungal communities

Across the diversity indices evaluated, the FM amended treatments significantly reduced the diversity (Shannon and Simpson) (Figs. 5C, 5D) and richness matrices (Chao1) (Fig. 5B) based on the observed number of OTUs. The least bacterial diversity across these indices was observed in the FM3 treatment. According to the ACE estimator, the FM3, while having the least diversity, was significantly similar to all other treatments ($p < 0.05$) (Fig. 5A). No significant difference ($p < 0.05$) was observed in the fungal community richness indices, ACE (Fig. 5E), Chao1 (Fig. 5F), and Shannon (Fig. 5G) among the treatments. However, Simpson's diversity index showed that FM2 had the least fungal diversity, while FM1 gave the highest diversity (Fig. 5H).

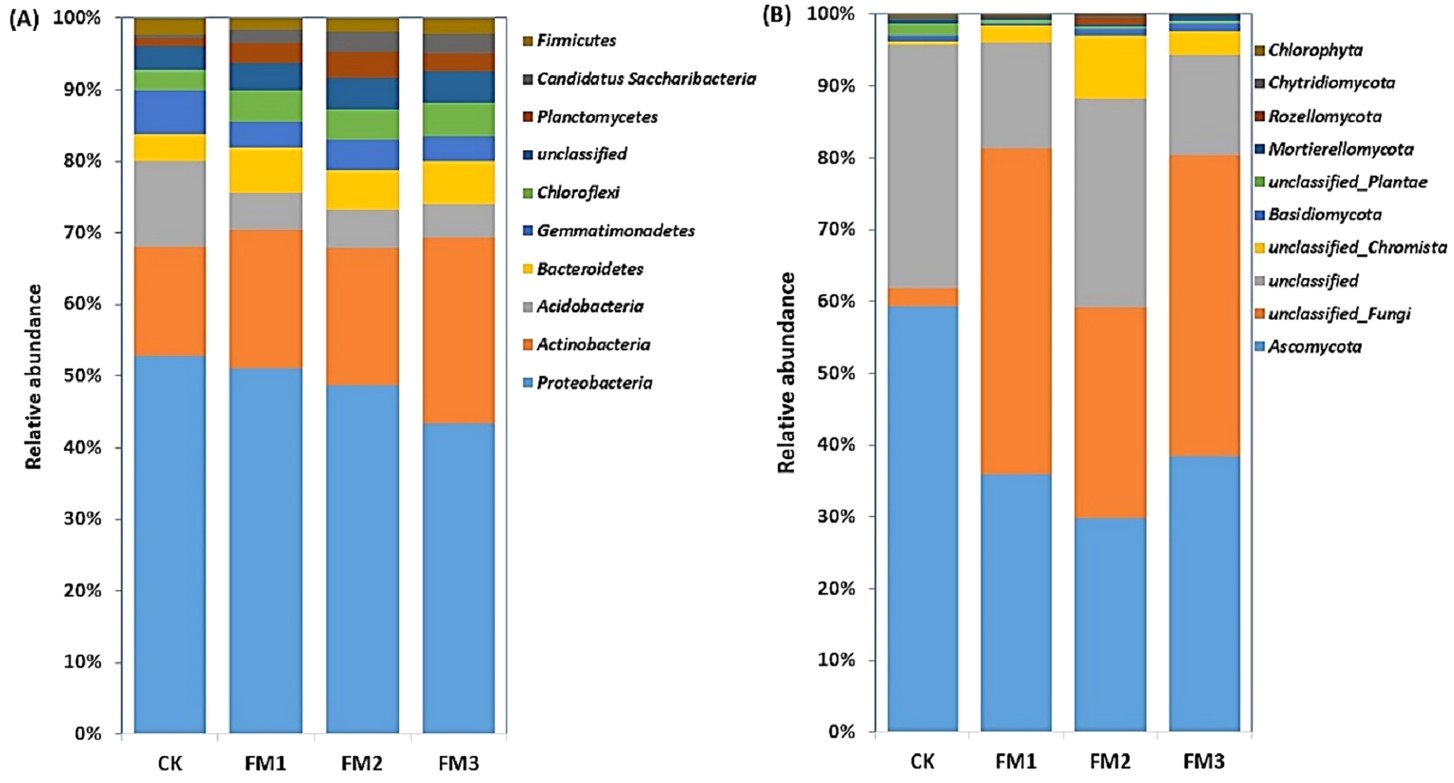

**Figure 3 Relative abundance of soil (A) bacterial and (B) fungal community at phylum level.** CK (control), FM: Filtered mud, FM1 (FM:soil at 1:4), FM2 (FM:soil at 2:3), FM3 (FM:soil at 3:2).

The beta diversity analyses represented by the principal coordinate analysis (PCoA) (Figs. 6A, 6B) showed the shift of bacterial and fungal community composition under different FM:soil ratios. Beta diversity among the soil samples was determined based on the unweighted Unifrac distance matrices. The PCoA of the bacterial taxa showed that the bacterial communities in FM1, FM2, and FM3 soil samples were clustered and separated from the CK soil sample. This indicates that the FM amendment significantly changed the soil microbial composition (Figs. 6A, 6B).

Similarly, the PCoA of the fungal taxa revealed an indistinct clustering pattern among the treatments (Fig. 6B). It was, however, observed that the control treatment was distinct from FM treatments (Fig. 6B). Distinct clustering of fungal communities in FM2 and FM3 was also observed in the PCoA.

## Relationship between soil properties and the distribution of bacterial and fungal communities

Redundancy analysis (RDA) was performed to explore the influence of soil variables on the relative abundance and community composition of bacterial and fungal phyla (Figs. 7A, 7B). It was observed that RDA1 and 2 accounted for 67.07% and 12.32%, respectively, for the changes in the bacterial phyla composition, while RDA1 and 2 accounted for 42.14% and 11.18%, respectively, for the differences in fungal community composition (Fig. 7B). The content of soil AP, pH, TOC, and TON positively influenced
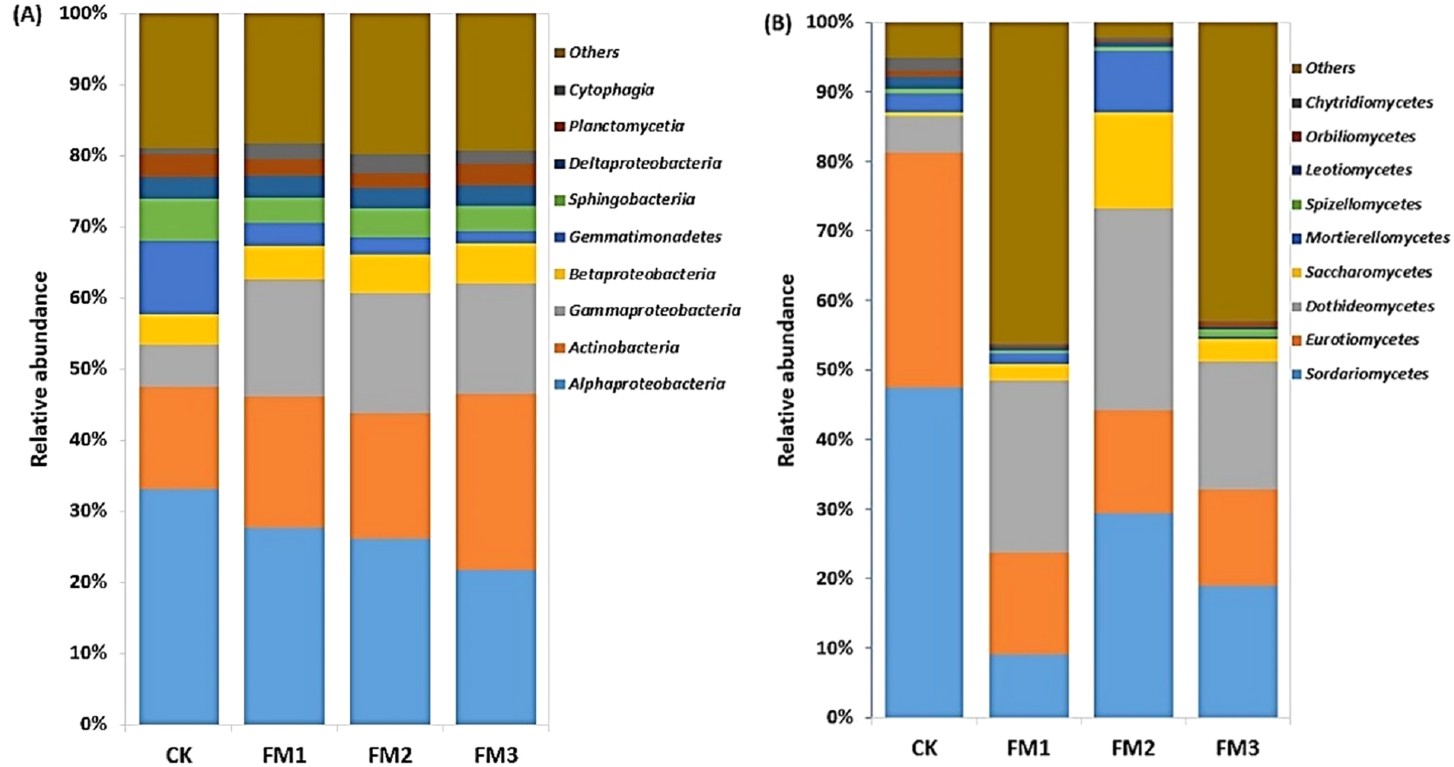

**Figure 4 Relative abundance of soil (A) bacterial and (B) fungal community at class level. CK (control), FM: Filtered mud, FM1 (FM:soil at 1:4), FM2 (FM:soil at 2:3), FM3 (FM:soil at 3:2).** CK (control), FM: Filtered mud, FM1 (FM:soil at 1:4), FM2 (FM:soil at 2:3), FM3 (FM:soil at 3:2).

the abundance of *Bacteroidetes* and *Chloroflexi*, while the availability of $NH_4^+$ and TON correlated with the relative abundance of the *Actinobacteria*. The concentration of soil $NO_3^-$ showed a close association with *Acidobacteria* and *Gemmatimonadetes*. Similarly, the availability of $NO_3^-$ influenced the abundance of *Ascomycota* (Fig. 7B). Likewise, TN, TON, pH, C/N ratio, and AP were the significant factors that influenced the abundance of *Basidiomycota*. The TOC of the treatments influenced the abundance of *Rozellomycota*.

## DISCUSSION

Organic amendments have been utilized as a substitute to inorganic fertilizers for enhancing soil fertility, crop yield, and quality (*Melero et al., 2006*; *Bonilla et al., 2012*). The improvements in the chemical properties of the soil due to the addition of FM could be attributed to its ability to retain soil nutrients and ensuring their slow release, in addition to its high nutritional contents. This was evident in the increase in plant height, number of tillers, and plant diameter under FM, especially in FM1. The observed increase in soil chemical properties due to FM is consistent with the finding of *Tayyab et al. (2018a)* who documented that the application of sugarcane straw to the soil improved its physicochemical properties. Therefore, the bioavailability of nutrients, especially the slow release of N, could be responsible for the improvement in some agronomic properties of sugarcane as observed in the FM treatments. Such improvement in wheat growth characteristics due to FM has been previously reported by *Khan, Khan & Zia (2012)*.

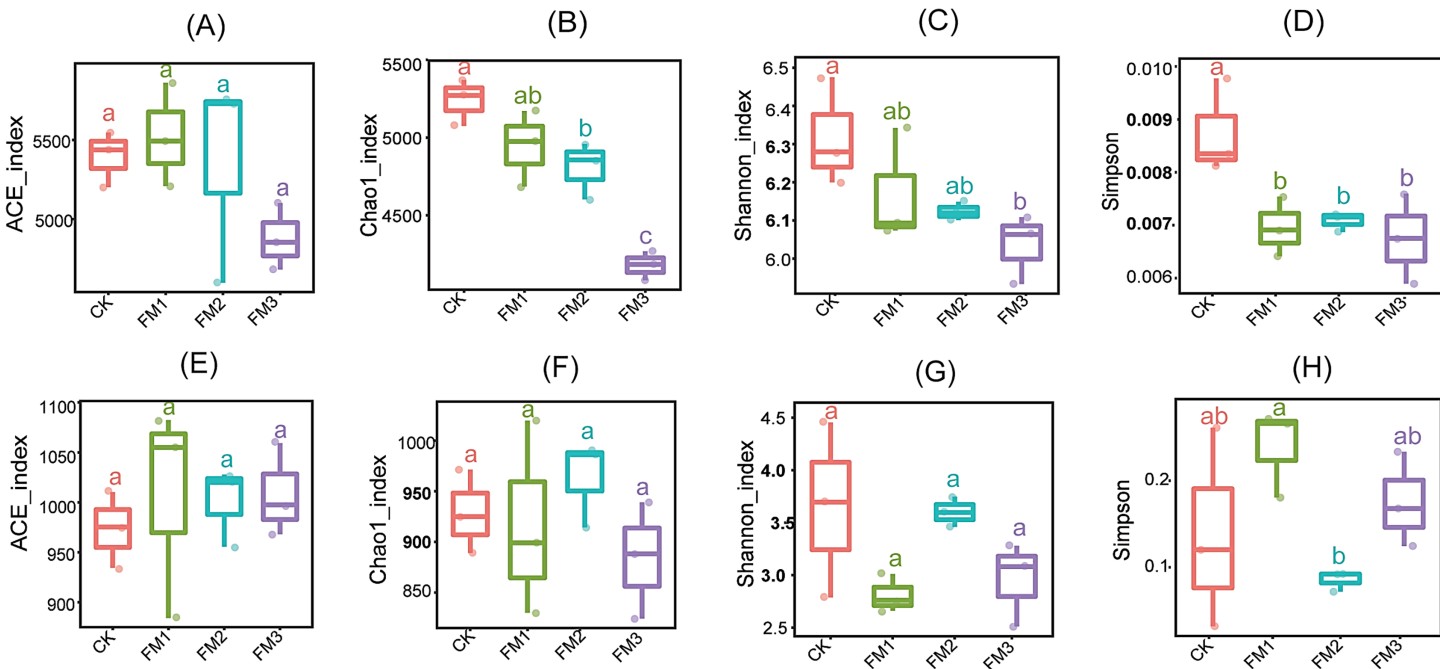

**Figure 5 Alpha diversity indices of bacterial (A-D) and fungal (E-H) community: (A, E) ACE, (B, F) Chao1, (C, G) Shannon, (D, H) Simpson. Error bars on the boxplots with different lowercase letters show significant differences between treatments (Tukey's test, [i]).** Error bars on the boxplots with different lowercase letters show significant differences between treatments (Tukey's test, $p < 0.05$). CK (control), FM: Filtered mud, FM1 (FM:soil at 1:4), FM2 (FM:soil at 2:3), FM3 (FM:soil at 3:2).

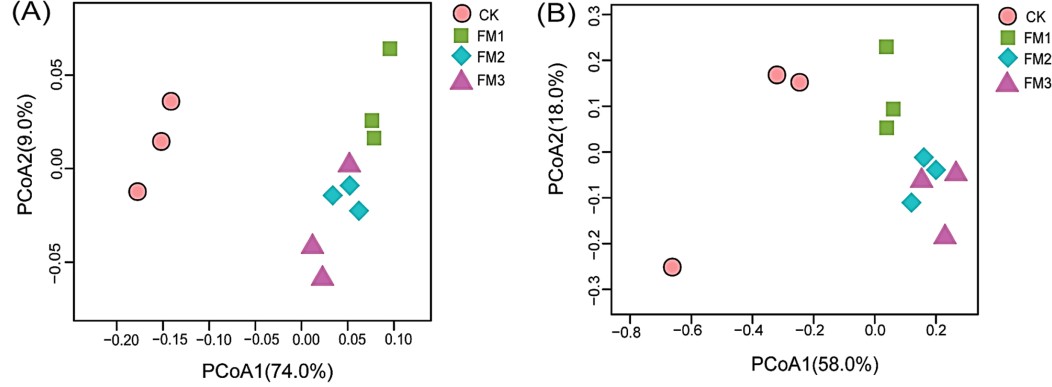

**Figure 6 Principal coordinate analysis (PCoA) of (A) bacteria and (B) fungi communities across the FM amended treatments. CK (control), FM: Filtered mud; FM1 (FM:soil at 1:4), FM2 (FM:soil at 2:3), FM3 (FM:soil at 3:2).** CK (control), FM: Filtered mud; FM1 (FM:soil at 1:4), FM2 (FM:soil at 2:3), FM3 (FM:soil at 3:2).

Several reports have also documented a positive effect of FM on productivity parameters and yield of several crops (*Yadav, 1992*; *Singh et al., 2003*; *Jamil, Qasim & Zia, 2008*; *Kumar et al., 2017*). Therefore, the improved agronomic characteristics of sugarcane as observed could also be attributed to the significant increase in soil nutrients, which were enhanced by the additional nutrient content of FM, and slow release of N. The application of organic amendments to the soil such as compost, cover crops, and manure has been

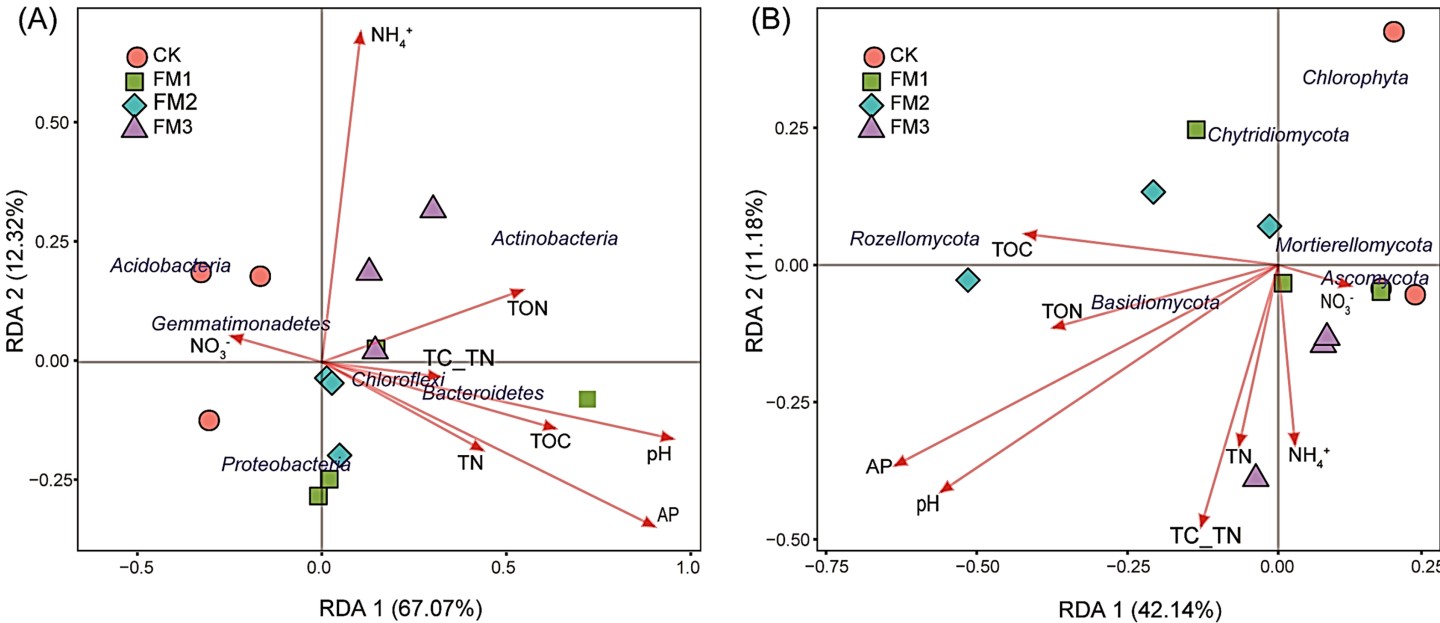

**Figure 7 Redundancy analysis (RDA) of the relative abundance of dominant (A) bacteria and (B) fungi community and soil environmental variables across the treatments. The length of soil variables arrow shows the relationship strength.** The length of soil variables arrow shows the relationship strength between the soil variables and the overall soil microbial community. CK (control), FM: Filtered mud, FM1 (FM:soil at 1:4), FM2 (FM:soil at 2:3), FM3 (FM:soil at 3:2); TN, Total nitrogen; TOC, Total organic carbon; TON, Total organic nitrogen; AP, Available phosphorous; TC_TN, Total carbon-Total nitrogen ratio.              

shown to enrich soils with slowly released available mineral nitrogen for plants use (*Bulluck et al., 2002*; *Cesarano et al., 2017*; *Ibrahim et al., 2020b*). Therefore, as an organic amendment derived from sugarcane, the use of FM does not only improves soil physical and chemical properties for sugarcane growth (*Rakkiyappan et al., 2001*) but also serves to return some of the nutrients taken up by the harvested sugarcane back into the soil, and thus, promoting nutrient cycling. Among the FM treatments, FM1 contained a higher amount of $NH_4^+$ which is the most needed N form for sugarcane growth. Unlike most other crops, evidence has shown that sugarcane has a preferential uptake of ammonium than other forms of N (*Boschiero, Mariano & Trivelin, 2018*). Although $NH_4$ concentration was similar in the CK, FM2, and FM3, the lower TN in CK suggests that higher mineralization of the soil's native N May have occurred, which may result in inorganic N losses if a lower uptake accompanied by leaching losses occurs. The reduction in $NO_3^-$ in FM treatments compared to the CK may have served to ensure a slow release to meet up the plant's needs. Organic amendments, including FM, are also known to immobilize inorganic N and ensure its slow release, thereby avoiding losses by leaching, runoff, or volatilization (*Kumar et al., 2017*; *Ibrahim et al., 2020b*).

Soil enzyme activities are essential indicators of soil fertility and are actively involved in nutrient cycling (*Nannipieri et al., 2012*). The increase in C and N cycling enzyme activities with an increase in FM rates showed that FM could be beneficial in C and N turnover and soil fertility. This was evident in the improved soil nutrient contents and growth of the sugarcane plant in the FM-treated soils. The increase of urease activity with increasing FM

rates as observed agrees with the report that its activity increased after organic fertilization (*Sardans, Peñuelas & Estiarte, 2008*; *Chen et al., 2017*). This may be linked to the increase in bacterial taxa linked to N-cycling in these treatments (Fig. 3A). Urease is required for the hydrolysis of urea (*Ibrahim et al., 2020a*). This was important for the mineralization of N present in the FM treatments for crop use. The increase in urease was associated with a slow release and availability of inorganic N in the FM treatments. Soil cellulase is responsible for the breaking down of cellulose (*Averill & Hawkes, 2016*). However, the reduction in its activity in FM treatments, especially in FM1 and FM2, could be linked to its inhibition of cellulase activity by reducing the abundance of microorganisms that act on cellulose, as observed in the reduction in the *Ascomycota*, which contain a wide range of cellulase degrading members (Fig. 3B). In addition, the increase in β-glucosidase in all the FM amended soil samples might be due to a rise in the labile organic C due to the increase in the total organic C provided by FM needed for its activity, as previously reported (*Simarani, Azlan Halmi & Abdullah, 2018*). β-glucosidase is an extracellular enzyme that helps in the mineralization of carbon (*Brookes et al., 2008*). Therefore, a higher amount of this enzyme in the FM treatments would be associated with the mineralization of organic C in the amendment and the associated availability of its mineralized form for other soil microbes involved in nutrient cycling.

The reactions of the soil microbial community to agricultural management are extremely complicated. Environmental factors such as moisture and soil temperature, which are often variably influenced by organic amendments, can regulate the diversity and community composition of soil organisms (*Carey et al., 2015*; *Vickers, 2017*). As an organic amendment, FM could effectively increase soil pH, drainage and stimulate soil microbial activities (*Tandon, 1995*; *Yang et al., 2013*). Some *Proteobacteria* members like the *Betaproteobacteria* are referred to as copiotrophic, and thrive in an environmental condition with a high C content and other nutrients (*Fierer, Bradford & Jackson, 2007*; *Yang et al., 2019*; *Ibrahim et al., 2020a*). This could explain their increased relative abundance in the FM treatments compared to the control (CK). The *Acidobacteria* had a higher relative abundance in the CK compared to FM-treated soils. This phylum is often found in low fertility soil, which explains its abundance in the CK treatment. Similar to our observations, *Acidobacteria* had a negative correlation with soil pH, as previously reported by *Lauber et al. (2009)* and *Rousk et al. (2010)*. Generally, *Acidobacteria* is referred to as Oligotrophs that grow successfully in natural cropland ecosystems (*Pershina et al., 2015*) and have been reported to contribute to the degradation of recalcitrant organic compounds (*Fierer et al., 2012*). Similarly, *Bryant & Frigaard (2006)* considered the *Chloroflexi* taxa as anoxygenic phototrophs, which play an essential role in the process of nitrification. However, the lower nitrification in the FM-based treatments where they were slightly increased may indicate that their taxonomic members present were not actively involved in the nitrification process.

The *Actinobacteria* was observed as one of the dominant bacterial phyla in the FM amended treatments, especially in FM3. Reports have also shown *Actinobacteria* to be prevalent in soils amended with press mud (filtered mud) (*Yang et al., 2013*). However, some contrary reports have shown that the typical soil has a higher relative abundance of

*Actinobacteria* than organically amended soils (*Francioli et al., 2016*; *Das et al., 2017*). Moreover, many members of the *Actinobacteria* phyla play important roles in N mineralization (*Fallah et al., 2021*; *Ibrahim et al., 2020b*), organic material decomposition like cellulose and chitin (*Ibrahim et al., 2021*), and helps in agrochemical degradation (*Bonanomi et al., 2016*). Additionally, the *Actinobacterial* taxa play a vital role in the transportation of soil phosphorous and biological control and have been referred to as beneficial microbes that improve agricultural soil quality (*Mander et al., 2012*). This could explain its positive correlation with soil AP, AK, and EC in the redundancy analysis. The relative abundance of the phylum *Firmicutes* was observed to increase with increasing FM proportions. This bacterial phylum flourishes in high C-rich soil and plays a crucial role in the degradation of complex organic materials (*Pershina et al., 2015*; *Tayyab et al., 2018b*), and is involved in the depolymerization of lignin (*Wu & He, 2013*). Filtered mud, which contains a proportion of the original sugarcane biomass, contains about 9% lignin (*Saleh-e-In et al., 2012*). This may explain the increase in abundance of the *Firmicutes* with increasing organic C sources due to higher FM proportions. The *Alphaproteobacteria*, which are non-nitrifying, survive on a small amount of soil nutrients and are the most dominant class of *Proteobacteria* in soils (*Yang et al., 2013*). This could explain their abundance in the CK treatment compared to the FM treated soil with higher nutrients. The *Betaproteobacteria* and *Gammaproteobacteria*, which contained the ammonium-oxidizing genera, *Luteimonas* and *Povalibacter*, were stimulated in the FM compared to the control treatments. Their presence was important to ensure gradual nitrification and the slow release of inorganic N from the abundant total N in the FM for plant use over time. *Luteimonas*, for example, isolated from rhizosphere soils was found to play important roles in N cycling (*Cheng et al., 2016*).

Among the fungal phyla, we observed that the phylum *Ascomycota* had a higher relative abundance across the treatments evaluated. Its reduction in FM compared to the control was associated with a reduction in its class members, *Eurotiomycetes* and *Spizellomycetes*. The reduction may be attributed to the decrease in the cellulase activity in FM treatments, as cellulase enzyme activity has often been associated with an abundance of fungi involved in the degradation of organic C sources (*Ibrahim et al., 2020b*). In line with our observations, *Lienhard et al. (2014)* and *Langarica-Fuentes et al. (2014)* reported that *Ascomycota* is the most common fungal phylum in the agroecosystems. The abundance of the fungal class *Dothideomycetes*, a member of the *Ascomycota* phylum in FM treatments, signified that they were the taxa that are actively stimulated under FM amendment for the degradation of organic matter introduced in the FM amendment.

In addition, the observed reduction in bacterial species richness and diversity in the FM treatments compared to the control treatment may have resulted from the stimulation and dominance of organisms involved in nutrient cycling, which outcompeted other organisms. The dominance of organisms involved in specialized functions in organic amended soils could increase their abundance and hence, reduce the diversity of other outcompeted organisms (*Ibrahim et al., 2020b*). The higher microbial diversity in the CK where there was lower nutrient composition could be associated with a higher community of oligotrophs (organisms that occasionally exist in soil with a low level of nutrients)

(*Fierer, Bradford & Jackson, 2007*). The clustering pattern of the FM amended treatments may indicate that there was an increase in soil nutrients, hence, a conducive environment for the dominance of bacterial and fungal communities involved in nutrient cycling, as observed in the increase in some nutrient cycling enzyme activities. This was evident in the enrichment of the *Actinobacteria, Gammaproteobacteria, Betaproteobacteria*, which contains general that are actively involved in N-cycling, as well as the organic C decomposing *Dothideomycetes*.

Microbial community composition shifts are regulated by soil physicochemical properties (*Zhalnina et al., 2015*; *Fallah et al., 2021*). As observed in our experiment, soil pH serves as one of the most important driving factors that regulate microbial diversity (*Bartram et al., 2014*; *Zhalnina et al., 2015*). Moreover, studies conducted by *Lauber et al. (2009)* and *Shen et al. (2013)* observed that soil pH in the natural environmental system is the most crucial factor in determining soil microbial communities' structure as it influences the availability of nutrients in the soil. Similarly, the nutritional composition of the soil is innately connected with the shifts in microbial distribution (*Zhang et al., 2019*; *Ibrahim et al., 2020a*, *2020b*). This was evident in our study as the effect of FM on soil nutritional properties was among the major factors the regulated the distribution of soil microbial populations.

## CONCLUSION

The evaluation of different FM:soil proportions on soil chemical properties, sugarcane growth, and the changes in the functional abundance of soil bacterial and fungal taxa was carried out. The use of FM improved the soil's chemical properties and the slow-release of $NO_3^-$. Specifically, FM1 increased the concentration of $NH_4^+-N$, the N fraction preferably taken up by sugarcane, which was associated with an increase in the plant height, and more improved growth properties, among other treatments. An increase in the proportion of FM also increased the activity of soil nutrient cycling enzymes. The use of FM reduced the diversity of soil bacteria while having an insignificant effect on fungal diversity. Its use stimulated the abundance of the bacterial phyla *Actinobacteria, Bacteroidetes, Acidobacteria*, and *Chloroflexi*, which are beneficial organisms, and the fungal phylum *Ascomycota* involved in degrading the lignocellulose complex.

The distribution of the soil microbial community under FM rates was regulated by the changes in soil pH and the availability of soil nutrients. Therefore, FM1 could be sustainably used in sugarcane production in clay-loam soils, as it is also the most practically applicable proportion, especially under field conditions. Future studies may consider the evaluation of the effects of FM on plant nutrients uptake, especially N, using isotopic tracer techniques in different soil types and under varied conditions.

## ACKNOWLEDGEMENTS

We appreciate the kind contributions of the team members of the Key Laboratory of Sugarcane Biology and Genetic Breeding, Fujian Agriculture and Forestry University, Fuzhou, China, for their immense contributions to the success of this research.

### Funding

This work was supported by the earmarked fund for China Agriculture Research System (No. CARS-17). The funders had no role in study design, data collection and analysis, decision to publish, or preparation of the manuscript.

### Grant Disclosures

The following grant information was disclosed by the authors:
China Agriculture Research System: CARS-17.

### Competing Interests

The authors declare that they have no competing interests.

### Author Contributions

- Ahmad Yusuf Abubakar conceived and designed the experiments, performed the experiments, analyzed the data, prepared figures and/or tables, authored or reviewed drafts of the paper, and approved the final draft.
- Muhammed Mustapha Ibrahim conceived and designed the experiments, prepared figures and/or tables, authored or reviewed drafts of the paper, and approved the final draft.
- Caifang Zhang performed the experiments, prepared figures and/or tables, and approved the final draft.
- Muhammad Tayyab analyzed the data, prepared figures and/or tables, and approved the final draft.
- Nyumah Fallah performed the experiments, prepared figures and/or tables, and approved the final draft.
- Ziqi Yang performed the experiments, prepared figures and/or tables, and approved the final draft.
- Ziqin Pang performed the experiments, prepared figures and/or tables, and approved the final draft.
- Hua Zhang conceived and designed the experiments, authored or reviewed drafts of the paper, and approved the final draft.

### Data Availability

The raw sequence files are available at the Sequence Read Archive: PRJNA755433; SAMN20822645 to SAMN20822648 for soil bacteria; SAMN20822649 to SAMN20822652 for the fungi.

### Supplemental Information

Supplemental information for this article can be found online at http://dx.doi.org/10.7717/peerj.12753#supplemental-information.

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
