# Peer review of "Filtered mud improves sugarcane growth and modifies the functional abundance and structure of soil microbial populations"

_PeerJ, doi:10.7717/peerj.12753_

## Round 0.1 · original submission · Major Revisions

Two reviewers are concerned with the small number of replicates (three per treatment), so please address this by calculating statistical power, confidence intervals, etc. Also, please be more specific about the communities and taxonomic entities you assessed, and address the remaining comments made by all reviewers.

Reviewer 1 ·

Basic reporting

In general, the writing and presentation of the data and information provided from the results of this study is quite good, but the information about the importance of using filtered mud is not clearly legible. The observed variables are also quite good with the data display informative enough. Something new from filtered mud has not appeared much in the discussion.

Experimental design

The experimental design of this study is not following statistical rules. Where if there are 4 treatments, accordingly, it should be at least 5 replications or 5 groups (in the case of RCBD). The way to determining replication is based on the formula t(n-1) > 15. While in this study the replications were only 3. As a result, the conclusions of this study are not valid according to design error.
The reasons for making comparisons in treatment are also not explained so that if implemented in the field, it is somewhat less suitable.

Validity of the findings

The design used does not meet the correct rules, because experimentally design, that determining the test or group is based on the formula t(n-1) > 15, so the minimum number of tests or groups used is 5, so statistically this result is not yet can legitimately conclude the results. The reason for using comparison treatments as in this study must have a basis. Because when applied in the field, it is very difficult to apply 3 parts of filtered mud with 2 parts of soil. Even though research in greenhouses is in pots, the implementation must also be considered when applied. And must also follow the rules of research. How to grow sugarcane in pots is also not explained. More details can be seen in the annotated manuscript.
The results of the study have been displayed well, but they are still often blocked because the tests used are not enough. The discussion is often less relevant and extends to organic materials in general, even though filtered mud of sugar cane has distinctive and different properties from organic materials used by other researchers. In general, a lot of discussions refer to the effect of the application of organic matter to the soil, there is no specificity of the organic matter you add. Why FM2 organic matter is lower has not been widely discussed, why, maybe because the sample is less representative. Moreover, how is it related to the higher FM content, the enzyme activity increases, while the plant performance is the opposite. The inconsistency of the results of diversity indices may be due to improper design.

Additional comments

In addition, the discussion is not in-depth and refers to the treatment material used so that the discussion extends to other matters which are quite unrelated to the material discussed in this study.

Annotated reviews are not available for download in order to protect the identity of reviewers who chose to remain anonymous.

Reviewer 2 ·

Basic reporting

In this study the authors assessed the impact of filtered mud (FM) amendment on soil chemistry, soil microbial community composition and sugarcane plant growth in greenhouse pot experiments. Three FM:soil mix ratios were compared against non-FM soils. The authors found that FM significantly altered soil mineral content, microbial community composition and improved several aspects of plant growth (height, tiller count, diameter).

The manuscript is well written and easy to read, and so is the research question straightforward and simple to understand. However, many aspects related to the microbial community analysis can be improved such as by using genus level units instead of limiting the analysis to phyla and classes, and by showing whether the FM material itself contained distinct consortia of microorganisms. These are likely to help with discussions as I found that the reported microbial taxonomies were too broad to infer their functions.

Experimental design

A major limitation of this study is the small number of replicates (three per treatment). Soil microbial communities are arguably one of the most complex ecosystems, and as such three replicates may not sufficiently capture the variability in this system (the overlapping data points in figure 5 for example).

In the microbial community analysis the authors do not go beyond phylum and class level lineages. As such, I found that the inferences of microbial function in these systems were overly broad and generalized. For example, authors discuss the importance of Actinobacteria, Firmicutes and Proteobacteria (lines 401 to 423) in soils, however, do all soil Actinobacteria, Firmicutes and Proteobacteria possess these functional capabilities? Authors should use at least genus level units to provide better resolution in identifying the exact microbial taxa responding to FM amendment.

Validity of the findings

I have several comments/questions related to the interpretation of the data:

Can authors provide reasons how FM increases plant productivity? Since FM is composed of organic matter, one assumption would be that plants are receiving more minerals from the FM amendment. Was mineral content in plant material measured to support this inference?

Is FM sterile or does it contain its own consortia of microorganisms? Is it possible that FM amendment introduced microorganisms into the system thus the distinct microbial community composition in FM soils compared with control? Are any of these microbial taxa potentially plant growth promoting species? If FM material contains beneficial or pathogenic microorganisms, how can we ensure these consortia are consistent across FM batches?

The snapshot data reported here is insufficient to show that FM slowed the release of N to match plant uptake and thus improve plant growth (e.g. line 343, 372). Were more samples taken serially to support this?

What are the reasons FM is preferred or superior to say biochar or other forms of organic amendment? Is there data comparing the benefits of FM and plant performance in FM to other amendments?

Additional comments

Line 187: Vernier caliper

Line 222: please describe what the sequencing library layout was, paired end vs single end, how many bp.

Line 227: which version of QIIME did authors use?

Line 227-235: Can authors provide an explanation as to why 97% OTUs were generated as opposed to using amplicon sequence variants (ASVs)? The sequencing library and platform used here should support ASVs. Please see https://www.nature.com/articles/ismej2017119

Line 235: how were microbial taxonomies inferred, with reference to which database?

Line 276-290: please provide statistical support for the reported enrichment/depletion of microbial and fungal taxa.

Line 294: class level lineages are too broad to make claims such as “ammonium oxidizing Betaproteobacteria and Gammaproteobacteria”. If would be better if authors can identify which ammonium oxidizing species within the beta and gammaproteobacteria classes are enriched in FM treatment.

Line 296: similar with above comment- which member of the Actinobacteria class are N-cycling?

Line 306: typo “choa”.

Line 332: typo “Choloflexi”

Line 349: “since a corresponding higher N uptake was observed in these treatments”. Can authors clarify which data indicates higher N uptake, was mineral content of plant material measured?

Line 373: “However, the reduction in cellulase activity in FM treatments, especially in FM1 and FM2, could be linked to its excessive utilization by the cellulolytic organisms to degrade the complex cellulose contained in the FM.” This is an interesting inference of the data- does the use of cellulase reduce its availability? Can authors comment on alternate explanations of the data e.g. FM inhibits cellulase activity and/or microorganisms that produce cellulase?

Line 377: is “unstable” carbon standard terminology?

Line 397-400: can authors please clarify how an increase in nitrifying bacteria (chloroflexi) is “a reason for the lower nitrification observed in the FM treatments”?

Line 424: this inference is false. Having the largest number of OTUs does not make Ascomycota the most abundant phyla (e.g. the OTUs are all low relative abundance).

Line 438-440: can authors clarify how clustering patterns in soil community composition indicate stimulation of beneficial microbial taxa? How is this inferred as there is no information relating to which exact taxa were enriched/depleted in these soils?

Sequence data not publicly available. A submission should be made to NCBI SRA and the BioProject accession number reported in the main text.

·

Basic reporting

I reviewed the article: "Filtered mud improves sugarcane growth and modifies the functional abundance and structure of soil microbial populations" for Peer J journal.

The article is clear and the English level is satisfactory. The authors evaluated the impact of the insertion of FM (filtered mud) on sugarcane development and on soil microbial communities. I have few comments about the paper.

It is important to highlight if the authors are evaluating rhizosphere or bulk soil samples. This definition could improve the discussion section. Furthermore, in the Results and Discussion sections (lines 294, 295, and 369 respectively) the authors talked about the enrichment of ammonia oxidizers and N-cycling organisms. However, they didn't evaluate specifically these communities. They concluded this because they observed a fluctuation in the relative abundance of Proteobacteria and Actinobacteria phyla.
If we consider the whole functional diversity associated with both phyla, It's almost impossible to affirm this. I suggest to the authors modify this part of the article.

Experimental design

The experimental design wasn't so innovative but it was efficient. I suggest to the authors briefly describe the methodology applied to evaluate the enzymatic activity.

Validity of the findings

The article brings interesting results, but part of these results was expected. But they are robust and controlled.

---

## Round 0.2 · Minor Revisions

Both reviewers seem happy and have accepted your manuscript. I only have one final request: Please include the R code used to run statistical analyses, either as supplementary material or as an open-access repository (like GitHub).

Reviewer 1 ·

Basic reporting

The authors have responded and improved according to my direction and recommendations and they have worked hard to improve this manuscript.

Experimental design

The author has defended his argument by showing statistical books and other explanations for experimental design which I recommend

Validity of the findings

After the author has corrected all the suggestions from the reviewers, this article is enough to add novelty to science and new discoveries from the object of his research

Additional comments

The research adds new information about microbial diversity and population due to mud fillers applied to soil. It has been written well enough so that readers will easily understand it and add scientific insight to other researchers.

·

Basic reporting

The authors worked on a new version of the article "Filtered mud improves sugarcane growth and modifies the functional abundance and structure of soil microbial populations" and they made some improvements to the text.
I was satisfied with these modifications.

Experimental design

Ok.

Validity of the findings

It is not a completely new article, but it was well conducted.

---

## Round 0.3 · accepted · Accept

Happy to accept your manuscript now, thank you for addressing my previous comments.